# Analysis of Melt Front Behavior of a Light Guiding Plate during the Filling Phase of Micro-Injection Molding

**DOI:** 10.3390/polym14153077

**Published:** 2022-07-29

**Authors:** Wei-Chun Lin, Fang-Yu Fan, Chiung-Fang Huang, Yung-Kang Shen, Liping Wang

**Affiliations:** 1School of Dental Technology, College of Oral Medicine, Taipei Medical University, Taipei 11031, Taiwan; weichun1253@tmu.edu.tw (W.-C.L.); fish884027@tmu.edu.tw (F.-Y.F.); chiung0102@tmu.edu.tw (C.-F.H.); 2Department of Dentistry, Taipei Medical University Hospital, Taipei 11031, Taiwan; 3School of Pharmacy and Medical Sciences, and UniSA Cancer Research Institute, University of South Australia, Adelaide, SA 5001, Australia; liping.wang@mymail.unisa.edu.au

**Keywords:** light guiding plate, micro-injection molding, melt front, flow characteristics

## Abstract

When the size of a liquid crystal display (LCD) increases, the light guiding plate (LGP) as the main part of the LCD must adopt a wedge-shaped plate to reduce its weight (the thickness of the LGP decreases because of this) and guide the light to the LCD screen. Micro-injection molding (MIM) is commonly used to manufacture LGPs. During the filling phase of MIM, the entire entering polymer melt front of the LGP should reach the end of the mold cavity at the same time. In this way, there will be no shrinkage or warpage of the LGP in its subsequent application, but it is difficult for the wedge-shaped LGP to meet these requirements. Therefore, the authors hoped to investigate MIM process parameters to change this situation. Otherwise, the LGP is easily deformed during the manufacturing process. Flow characteristics of LGPs were investigated during the filling phase of MIM in this study. Experimental and 3D numerical simulations were used to analyze the hysteresis, i.e., the advance of the polymer melt front of the LGP in MIM. Study results showed that a low injection speed caused a hysteresis effect of the plastic melt front, the solution was to increase the injection speed to improve the situation and an injection speed of 10 cm/s could achieve uniformity of the melt front in MIM. The research results showed that the filling situation of the LGP of MIM in the experiment was very close to that of the 3D numerical simulation.

## 1. Introduction

The light guiding plate (LGP) is the main component of a liquid crystal display (LCD). It transmits light from a cold cathode fluorescent lamp (CCFL) to the LCD surface. In particular, the function of the LGP is to uniformly guide light to the LCD. The optical properties and scale quality of LGPs are extremely significant in LCD systems. When the size of an LCD increases, the LGP must be configured as a wedge-shaped plate to reduce its weight (therefore, the thickness of the LGP should also be reduced) and direct light to the LCD screen.

There are two principal methods for manufacturing an LGP: injection molding and hot embossing (or nano-imprinting). The advantage of injection molding is that its fabrication time is short, so the manufacturing speed is fast. With hot embossing, the LGP is less likely to warp and shrink due to planar pressure, but its manufacturing time is longer. Micro-injection molding (MIM) is a relatively new and rapidly evolving technology that can easily produce components of scale and detail which cannot be achieved by conventional injection molding techniques. MIM cannot simultaneously fill the wedge-shaped cavity of an LGP because the cavity has various thicknesses. The entire polymer melt front of the LGP should reach the end of the mold cavity at the same time during the filling phase of the MIM process, so that shrinkage and warpage of the LGP do not subsequently occur. However, when using a wedge-shaped LGP, this is difficult to achieve, and therefore the process parameters of MIM must be adjusted to change this situation. If not, the LGP will easily be deformed during the MIM process. Kukla et al. [1] demonstrated that MIM can be used when (1) the mass of a part is on the order of a few milligrams, (2) the dimensions of a part’s features are in the micrometer range, and (3) the part can have any dimensions, but its tolerances are in the micrometer range.

Yu et al. [2] revealed that the injection speed and mold temperature during injection molding markedly impact the replication accuracy of microstructures on metal mold inserts. Bűrkle and Wohlrab [3] demonstrated that an optical element, such as a lens or prism, has low birefringence, residual stress, and clamping force using injection compression molding. Yoshii et al. [4] analyzed the replication of micro-grooves on an injection-molded optical disk and reported that a high mold temperature increases the replication accuracy of micro-grooves. Su et al. [5] demonstrated that the mold temperature must be high, such that a polymer can easily fill the cavity. Heckele and Schomburg [6] compared the use of various polymers (such as cyclic-olefin copolymer (COC) and polymethyl methacrylate (PMMA)) with different mold methods (such as hot embossing, injection molding, reaction injection molding, compression molding, and thermoforming) to produce micro-micro parts. Chien and Chen [7] and Kuo and Su [8] studied the fabrication of V-cut patterns of LGPs. Li et al. [9] determined the light-scattering hemispheric microstructure of an LGP. Shen [10] analyzed the effect of process parameters on the height replication of microstructures on LGPs manufactured by MIM and micro-injection-compression molding (MICM). The mold temperature was the most significant process parameter with both MIM and MICM. Kim et al. [11] found that a higher luminance of the LGP was achieved with concave microlens arrays (MLAs) compared to convex MLAs. Park et al. [12] presented the design and fabrication of a light-emitting diode (LED)-based LGP to realize a diffuser sheet-less application for indoor lighting. Huang et al. [13] investigated the usefulness of induction heating to rapidly heat a mold, thus enhancing the replication effect of the microstructure of an LGP during injection molding. Hong et al. [14] developed rapid heat-cycle molding (RHCH) to obtain a high transcription ratio of the microstructure uniform thickness and low birefringence. Tsai et al. [15] showed that the mold temperature was the most important process parameter for microstructures of LGPs in injection molding.

Dong et al. [16] simulated MIM of LGPs and provided temperature and pressure distributions of LGPs during the filling phase. Chung et al. [17] fabricated a flexible LGP using CO_2_ laser lithografie, galvanoformung, abformung (LIGA)-like technology including laser-ablated microstructures of a PMMA mold and polydimethylsiloxane (PDMS) casting process. Compared to photolithography, knife tool machining, hot embossing and injection molding, Chung et al.’s methods had advantages of rapid fabrication, cheap equipment, and easy integration. Jung et al. [18] indicated that adopting high-speed injection molding for thin-walled LGPs provided advantages such as weight, shape, size, and a reduction of production costs. Wang et al. [19] developed a complete fabrication process of LGPs using inkjet printing technology. A high injection speed and high molding pressure were required to completely fill the mold, but it was easy to produce parts with defects similar to high residual stress and warpage. Kuo et al. [20] used injection molding to fabricate LGPs with microstructures. They wanted to make the depth and angle of the V-cut microstructures as close to the target values and minimize the LGP’s residual stresses. Yu et al. [21] analyzed the cylindrical micro-feature design of a front light LGP and concluded that the cylindrical convex micro-features arranged at the bottom surface of the LGP would provide an optimum effective contrast ratio. Jakubowsky et al. [22] designed, simulated, and manufactured LGPs with undercut microstructures for vertical light emission on PMMA material using a variothermal injection molding process. The experimentally determined illumination performance characteristics (LID) and predictions based on optical simulations revealed good agreement for the LGPs they produced.

Lee et al. [23] considered an inverse-trapezoidal micropattern (ITM) as an optimal light out-coupler for an LGP in display devices. They fabricated trapezoidal micropatterns using a rigid metallic mold and then bonded it to a flat LGP substrate without optical glue. Li et al. [24] examined the difference in density between the light guide point distribution of LGP images, with different sizes, shapes, and brightness levels of LGP defects, and limitations of a small number of defective samples. Hoang et al. [25] investigated interactions between a metal surface and multiple pulses to achieve optimal focusing conditions during ultrashort laser ablation. Li and Li [26] used automatic vision-based defect detection on an LGP based on the low contrast between the defect and the background, an uneven brightness, and a complex gradient texture. The experimental results showed that the F1-score on the two datasets reached 99.67% and 96.77%, respectively, which verified the effectiveness of the method. Quesada et al. [27] used micro-groove arrays to etch a glass LGP, transforming edge-lit incident light into a uniform area output from its surface. Luminance uniformities in excess of 80% for 9-point and 75% for 455-point measurements were achieved with a 1.2% extraction efficiency per groove. Wu and Lieu [28] used roll-to-roll technology to form optoelectronic products. The bottom surface of the composite LGP featured dot microstructures, while the upper surface featured prismatic microstructures. The proposed design for a composite LGP with a double-sided structure significantly increased the central average brightness and the uniformity of the luminance.

Morini [29] focused on the role of viscous heating in a fluid flowing through micro-channels. The effect of viscous heating became very important in liquid flows when the hydraulic diameter of the micro-channel was <100 μm. Hatzikiriakos and Roseubaum [30] developed a technique to determine the slip velocity at high shear rates corrected for the effect of viscous heating in capillary flow. Numerical simulations at the micro-scale combined with the macro-scale are helpful for injection molding. Zhou et al. [31] represented a dual-scale model, which coupled a macroscopic flow field of the film-casting process and the microscopic crystallization behavior of the material. The macro-scale results in terms of the width distribution and temperature variations and the microscale morphological results in terms of spherulitic and cylindrical crystal development were all well within the experimental data.

High-speed cameras are usually used to capture the flow of the melt front in the filling stage of injection molding. The process parameters (especially the injection speed and injection pressure) are used to improve and enhance the plastic melt front to reach the end of the mold cavity at the same time [32,33,34,35]. This is the first study to apply conservation equations (continuity, momentum, and energy) to calculate viscous heating with different thicknesses at different positions of the cavity, and then convert it into a temperature value to understand the temperature value at the melt front. Then, this study used the concept of viscosity to calculate viscous heating, to infer the reason why the melt front is uniform or hysteretic. We then changed the injection speed to achieve a melt front balance (that is, each point of the melt front reached the end of the mold cavity at the same time), and we finally reduced the warpage of the LGP during the injection molding process.

This study designed a novel gate and combined it with a suitable process to overcome inconsistencies in MIM. The main purpose of this study was to focus on the injection molding machine used to create the LGP. Changing process parameters of the machine and the theory (physics and mathematics) of the filling phase during the MIM process were used to overcome problems normally encountered when manufacturing LGPs. These concepts can be applied to all machines to improve the process of fabricating LGPs during MIM. Experimental and three-dimensional (3D) numerical simulations were used to analyze the hysteresis and advance the polymer melt front of the LGP in the MIM process. A theoretical analysis using fluid mechanics and heat transfer was applied to explain the hysteresis and advance of the polymer melt front to improve the filling situation of LGPs during MIM. This situation was optimized, that is, the entire polymer melt front was able to reach the end of the mold cavity at the same time.

## 2. Experimental Procedures

### 2.1. Equipment and Materials

These MIM experiments were carried out using an injection molding machine (CLF-125T, Taiwan Chuanlifu Co., Ltd., Zhongli, Taiwan). The tempering unit (the machine that produced the mold temperature for heat transfer (which used water)) was manufactured by Byoung Taiwan. Its range of control temperatures was 20–140 °C. A two-plate mold is commonly used in MIM studies. Figure 1 illustrates the layout and dimensions of the mold. The dimensions of the mold insert were 100 (length) × 75 mm (width). A wedge-shaped LGP was used in this study. The thick and thin ends of the mold insert were 2.8 and 0.8 mm thick, respectively. The microstructure pattern of the surface of the mold insert was a hemispherical shape. The microstructure was 36 μm high with a radius of 200–300 μm through linear expansion from the thick end to the thin end of mold insert’s thickness. The material of the mold insert was SUS 430 stainless steel. A wet etching method was used to make the steel mold insert. Figure 2 reveals the process chart for photo-etching of the mold insert. An LGP traditionally employs a transparent material. Injection-grade PMMA (Delpet 80NH, Asahi Kasei, Tokyo, Japan) was used for the material of the LGP in this study.

Three process parameters (mold temperature, melt temperature, and injection speed) were used to evaluate the effects of process conditions on the flow characteristics during the filling phase of MIM. A fan-gate with a cross-sectional area of 24.24 cm^2^ was applied for the mold cavity. Table 1 lists the MIM process parameters and their values. This study evaluated flow characteristics of LGPs surveyed during the filling phase of MIM. We judged two characteristics of the flow characteristics: melt front uniformity (flow uniformity) and melt front hysteresis (flow hysteresis) during the filling phase. The definition of melt front hysteresis is that the entire melt front does not simultaneously reach the end position of the LGP during the filling phase of MIM. The definition of melt front uniformity is that the entire melt front reaches the end position of the LGP at the same time during the filling phase of MIM.

### 2.2. Mathematical Model and Numerical Simulation Method

Mass, momentum, and energy conservation equations for non-isothermal generalized Newtonian fluids on MIM are as described here.

Continuity equation:(1)$$\frac{\partial \rho }{\partial t}+\frac{\partial (\rho u)}{\partial x}+\frac{\partial (\rho v)}{\partial y}+\frac{\partial (\rho w)}{\partial z}=0$$

Momentum equation:(2)$$\rho \left(\frac{\partial u}{\partial t}+u\frac{\partial u}{\partial x}+v\frac{\partial u}{\partial y}+w\frac{\partial u}{\partial z}\right)=-\frac{\partial P}{\partial x}+\eta \left(\frac{\partial^{2}u}{\partial x^{2}}+\frac{\partial^{2}u}{\partial y^{2}}+\frac{\partial^{2}u}{\partial z^{2}}\right)+\rho g_{x}$$
(3)$$\rho \left(\frac{\partial v}{\partial t}+u\frac{\partial v}{\partial x}+v\frac{\partial v}{\partial y}+w\frac{\partial v}{\partial z}\right)=-\frac{\partial P}{\partial y}+\eta \left(\frac{\partial^{2}v}{\partial x^{2}}+\frac{\partial^{2}v}{\partial y^{2}}+\frac{\partial^{2}v}{\partial z^{2}}\right)+\rho g_{y}$$
(4)$$\rho \left(\frac{\partial w}{\partial t}+u\frac{\partial w}{\partial x}+v\frac{\partial w}{\partial y}+w\frac{\partial w}{\partial z}\right)=-\frac{\partial P}{\partial z}+\eta \left(\frac{\partial^{2}w}{\partial x^{2}}+\frac{\partial^{2}w}{\partial y^{2}}+\frac{\partial^{2}w}{\partial z^{2}}\right)+\rho g_{z}$$

Energy equation:(5)$$\rho_{l}c_{pl}\left(\frac{\partial T_{l}}{\partial t}+u\frac{\partial T_{l}}{\partial x}+v\frac{\partial T_{l}}{\partial y}+w\frac{\partial T_{l}}{\partial z}\right)=\eta {\dot{\gamma }}^{2}+k_{l}\left(\frac{\partial^{2}T_{l}}{\partial x^{2}}+\frac{\partial^{2}T_{l}}{\partial y^{2}}+\frac{\partial^{2}T_{l}}{\partial z^{2}}\right)$$
(6)$$\rho_{s}c_{ps}\left(\frac{\partial T_{s}}{\partial t}\right)=k_{s}\left(\frac{\partial^{2}T_{s}}{\partial x^{2}}+\frac{\partial^{2}T_{s}}{\partial y^{2}}+\frac{\partial^{2}T_{s}}{\partial z^{2}}\right)$$
(7)$$\dot{\gamma }=\sqrt{{\left(\frac{\partial u}{\partial x}\right)}^{2}+{\left(\frac{\partial v}{\partial y}\right)}^{2}+{\left(\frac{\partial w}{\partial z}\right)}^{2}}$$

In these equations, *x*, *y*, and *z* are Cartesian coordinates; t is time; $\rho_{s}$ and $\rho_{l}$ are solid and liquid densities, respectively; *u*, *v*, and *w* are velocities; *P* is pressure; g is gravity; η is viscosity; $c_{ps}$ and $c_{pl}$ are solid and liquid specific heats, respectively; $k_{s}$ and $k_{l}$ are solid and liquid thermal conductivities, respectively; $T_{s}$ and $T_{l}$ are solid and liquid temperatures, respectively; $\dot{\gamma }$ is the shear rate; and $\eta {\dot{\gamma }}^{2}$ is viscous heating.

This study used the control volume finite element method (CVFEM) to solve Equations (1)–(7). Shen et al. [36] revealed details of the mathematical model and numerical simulation method. Figure 3 indicates the meshes of the model for the 3D numerical simulation. The 3D mesh in the MoldFlow (vers. 2009, San Rafael, CA, USA) analysis was utilized to examine flow characteristics of the LGP during the filling phase of MIM. The mesh type was a four-node tetrahedral element. There were 102,234 elements and 23,977 nodes on the meshes of the numerical simulation. Numerical simulations were performed using an IBM personal computer (PC). The central processing unit was a 3.5 GB Pentium 6. An 8-GB random access memory and a 1-TB hard drive were used in the PC. The calculation time of each case was 5 h and 37 min for the numerical simulation.

Figure 4 shows the position of the polymer melt front of the LGP during the filling phase of MIM (the process parameters were a mold temperature of 60 °C, a melt temperature of 245 °C, and an injection speed of 3 cm/s). The entire polymer melt front of the LGP did not reach the end of the mold cavity at the same time. This phenomenon caused shrinkage and warpage of the LGP after molding. The authors attempted to resolve this situation by a 3D numerical simulation. The authors used six melt front discussion points (P1–P6) in Figure 1 to discuss the melt front position. In Figure 4, the authors controlled the shape of the plastic melt front in the entire filling stage with different injection times during the filling process of the LGP during MIM. Therefore, the melt fronts of different shapes on the LGP can be seen in Figure 4. Placing the LGP on a checkered plate was used to confirm the melt front positions of the six discussion points in Figure 1. The situation (as shown in Figure 4) of the melt front shows hysteresis, which is not conducive to the molding process. We discuss how to improve this situation later in this study.

## 3. Results and Discussion

In this study, we first dealt with the bad flow situation of the LGP with MIM (Figure 4). Through experimentation and 3D numerical simulation, the authors obtained a bad flow situation. After the authors received the results of the bad flow situation, we wanted to resolve this situation by a 3D numerical simulation. A theoretical analysis of fluid mechanics and heat transfer was applied to explain the hysteresis or advance the ability of the polymer melt front to overcome the filling situation of the LGP during MIM. The situation was optimized; that is, we found a solution so that the entire polymer melt front reached the end of the mold cavity at the same time.

Figure 5 reveals the filling phase of the LGP with MIM between the experiment and 3D numerical simulation. The process conditions were a mold temperature of 60 °C, a melt temperature of 245 °C, and an injection speed of 3 cm/s. At the beginning of the filling phase, the melt front of the central region led the melt front of the sidewall region of the mold cavity. However, the melt front of the central region reached the middle of the cavity. Compared to the position of the melt front of the sidewall region, there was a flow hysteresis phenomenon (melt front hysteresis) at the 90% filling phase. At the end of the filling phase, the melt front of the sidewall region remained ahead of that of the central region. Finally, the center area of the mold cavity was filled. Therefore, it is necessary to discuss melt front hysteresis in the central region of the mold cavity (Figure 5). Previous experimental results were the most undesirable results in this study. Our research goal was to overcome this flow hysteresis phenomenon. To do this, the authors attempted to increase the injection speed to resolve this situation with MIM. We theoretically analyzed various positions of the polymer melt front when filling the mold cavity with MIM (using the viscous heating term ($\eta {\dot{\gamma }}^{2}$) in the energy equation). The authors discussed the flow characteristics of the filling phase of MIM through a 3D numerical simulation (theory). Figure 6 shows the uniformity and hysteresis of the melt front during the filling phase with MIM. As the injection speed increased, the melt front hysteresis could be resolved by a 3D numerical simulation.

In this study, we attempted to understand the situation of the melt front uniformity and hysteresis during the filling phase of MIM. Next, the authors discuss the melt front uniformity and hysteresis by viscous heating, the temperature distribution, and velocity distribution during the filling process of the mold wall depth at various positions by a 3D numerical simulation.

Figure 7 shows the 30% filling phase of MIM according to the 3D numerical simulation (discussion points P1–P3 in Figure 5). Figure 7a,c,e show the viscous heating ($\eta {\dot{\gamma }}^{2}$), and temperature and velocity distributions under the process conditions of a mold temperature of 60 °C, a melt temperature of 245 °C, and an injection speed of 10 cm/s. In the process conditions of Figure 7b,d,f, only the injection speed changed to 3 cm/s, and the rest of the process parameters remained the same. The melt front distribution of the numerical determination was similar to that of the experimental determination (Figure 6). On the cavity surface, when the filling phase was 30%, the melt front of the central region still led the melt front of the sidewall region. The authors first discuss the melt front hysteresis at the 30% filling phase. Viscous heating was approximately the same at positions P1–P3 on the cavity surface (with thicknesses of 0.58 and 2.69 mm). However, viscous heating at position P2 inside the cavity (with thicknesses of 0.88, 1.19, and 2.28 mm) was less than that at positions P1 and P3. High viscous heating generates higher temperatures. The temperature distributions were generally the same at positions P1–P3 on the cavity surface (with thicknesses 0.58 and 2.69 mm), but temperatures at positions P1 and P3 inside the cavity (with thicknesses of 0.88, 1.19, and 2.28 mm) were higher than that at position P2. The high temperature of the polymer melt reduced its viscosity. Because of this, the flow resistance of the polymer melt decreased, and the melt front velocity increased. The velocity distribution of the central thickness at position P2 was smaller than those at positions P1 and P3. The velocity distribution of the central region was smaller than that of the sidewall region. We first discuss the melt front uniformity at the 30% filling phase. Discussing the position of the melt front in the central region is worthwhile in this study (Figure 6a). Figure 7a,c,e reveal the melt front uniformity for viscous heating ($\eta {\dot{\gamma }}^{2}$), and temperature and velocity distributions in the thickness direction of the mold cavity during the 30% filling phase of MIM (Figure 6a). Viscous heating was roughly the same at positions P1–P3 on the cavity surface (with thicknesses of 0.58 and 2.69 mm). However, viscous heating at position P2 inside the cavity (with thicknesses of 0.88 and 1.19 mm) was higher than those at positions P1 and P3. High viscous heating increased the polymer melt temperature. The temperature distributions were generally the same at positions P1–P3 on the cavity surface (with thicknesses of 0.58 and 2.69 mm). Yet, temperatures at positions P1 and P3 inside the cavity (with thicknesses of 0.88 and 1.19 mm) were lower than that at position P2. The high temperature decreased the viscosity of the polymer melt; thus, the flow resistance of the polymer melt decreased and induced an increase in the melt front velocity. The velocity distribution at position P2 was slightly larger than those at positions P1 and P3. In summary the velocity distribution in the central region of the mold cavity was slightly larger than that in the sidewall region.

When the injection speed increased, it was evident that the viscous heating at position P2 was greater than those at positions P1 and P3 with different cavity thicknesses. Therefore, the temperature at position P2 was higher than those at positions P1 and P3. The higher temperature decreased the viscosity of the melt polymer at position P2. At that time, the velocity of the melt polymer became faster at position P2. This was not evident from the MoldFlow analysis of the filling phase. Results of Figure 7a,c,e corrected the defect in the results of Figure 7b,d,f and the experiment shown in Figure 6b near the end of the filling phase (60% filling phase). In this study, we can fully understand the phenomenon of melt front uniformity and hysteresis. The results can serve as a reference for melt front uniformity and hysteresis at the 60% filling phase of MIM.

The authors analyzed the flow characteristics in the central region of the mold cavity (discussion points P4–P6 in Figure 5) during the filling phase of MIM. The right sides of Figure 8a–e show viscous heating, and the temperature and velocity distributions in the thickness direction of the mold cavity during the 60% filling phase of MIM for melt front hysteresis (Figure 6b), while the left sides of Figure 8a–e indicate the melt front uniformity. Viscous heating was roughly the same at positions P4–P6 on the cavity surface (with thicknesses of 0.58 and 2.69 mm). Viscous heating at position P5 inside the cavity (with thicknesses of 1.57 and 2.49 mm) was less than those at positions P4 and P6. High viscous heating increased the polymer melt temperature. The temperature distributions were generally the same at positions P4 and P6 on the cavity surface (with thicknesses of 1.25 and 2.69 mm). However, temperatures at positions P4 and P6 inside the cavity (and thicknesses of 1.6 and 2.0 mm) were higher than that at position P5. These high temperatures reduced the polymer melt viscosity. Hence, the flow resistance of the polymer melt decreased and the melt front velocity increased due to the lower flow resistance. The velocity distribution at position P5 was smaller than those at positions P4 and P6. The velocity distribution in the central region of the mold cavity was smaller than that in the sidewall region. The comparative results of positions P2 and P5 indicated that viscous heating of position P2 was higher than that of position P5 at the same thickness direction (with thicknesses of 1.5 mm). Thus, the temperature and velocity distributions at position P5 were smaller than those at position P2. The melt front of the central region slowly advanced during the filling phase of MIM. The left sides of Figure 8a–e show viscous heating and temperature and velocity distributions of the melt front uniformity during the 60% filling phase (Figure 6a). Viscous heating was approximately the same at positions P4–P6 on the cavity surface (with thicknesses of 1.28 and 2.69 mm). However, viscous heating at position P5 inside the cavity (with thicknesses of 1.68 and 1.79 mm) was less than those at positions P4 and P6. High viscous heating increased the temperature of the polymer melt. The temperature distributions were generally the same at positions P4–P6 on the cavity surface (with thicknesses of 1.28 and 2.69 mm). Temperatures were also almost the same at positions P4–P6 inside the cavity (with thicknesses of 1.68 and 1.79 mm). A high temperature reduced the polymer melt viscosity. Therefore, the flow resistance of the polymer melt decreased, and the velocity of the polymer melt increased. The velocity distribution at position P5 was slightly smaller than those at positions P4 and P6. The velocity distribution of the central region of the mold cavity was slightly smaller than that of the sidewall region.

The melt front of the central region of the mold cavity advanced to the sidewall region at the beginning of the filling phase of MIM, but the melt front of the central region was delayed compared to the sidewall region in the 60% filling phase. The reason was that the melt front passed through the center, and the thickness of the cavity rapidly decreased as the filling phase progressed, thereby increasing the flow resistance of the polymer melt. Therefore, the melt front velocity slowed. The melt front moved obliquely in the sidewall region. Flow routes of the polymer melt rapidly increased as the cavity’s thickness slowly changed. Therefore, the flow resistance of the polymer melt was reduced. The increased friction between the polymer melt and mold wall created an additional flow route for the melt front. Increased viscous heating of the polymer melt caused the polymer melt temperature to rise and the polymer melt viscosity to decrease. Based on this situation, the flow resistance of the polymer melt decreased. The velocity of the melt front increased, which caused the melt front to easily fill in the thin cavity, thereby increasing the filling velocity. As the injection speed increased, the friction of the melt front between the polymer melt and mold wall increased in the central region of the mold cavity. Viscous heating in the central and sidewall regions was almost the same. Since the increase in viscous heating led to an increase in the melt front velocity, this caused the melt front in the center and sidewall regions to simultaneously reach the end of the mold cavity at the end stage of filling. Previous results indicated the effect of the hysteresis distance on various process parameters during the filling phase of MIM. The hysteresis distance is the distance from the melt front of the central region to the mold wall. The hysteresis distance decreased as the injection speed increased.

Figure 9 shows the molded LGP by MIM. Suitable processing conditions were an injection speed of 10 cm/s, a mold temperature of 60 °C, and a melt temperature of 245 °C, and these conditions could fabricate the molded LGP well during the filling phase of MIM. Future research will discuss the luminance and brightness field distribution of the LGP under various MIM process parameters. The authors surmise that the mold temperature will be the most important processing parameter affecting the illuminance field distribution of molded LGPs fabricated by MIM from our laboratory experience. The molded LGP with microstructures had better uniformity than that without microstructures.

## 4. Conclusions

The purpose of this study was to determine the flow characteristics of a wedge-shaped LGP during MIM. The position of the polymer melt front was very close between the experiment and the 3D numerical simulation of MIM. In numerical simulations and experimental results, when using low injection speeds for MIM, the hysteresis of the melt front in the central region of the cavity was smaller than that on the two sides of the cavity. The analysis showed that increasing the injection speed could overcome the unbalanced filling situation (hysteresis of the melt front) in MIM. Results showed that the position of the melt front in the central area of the cavity was the same as the position on both sides of the cavity at the end of filling. As the injection speed increased, the friction of the melt front between the polymer melt and mold wall in the central region of the mold cavity also increased. Viscous heating in the center and sidewall regions was almost identical. Due to the increased viscous heating, the melt front velocity increased, causing the melt fronts in the center and sidewall regions to simultaneously reach the end of the cavity at the end of the filling phase of MIM. A theoretical analysis (the concept of viscous heating) was used to explain the filling situation of the LGP during MIM.

## Figures and Tables

**Figure 1 polymers-14-03077-f001:**
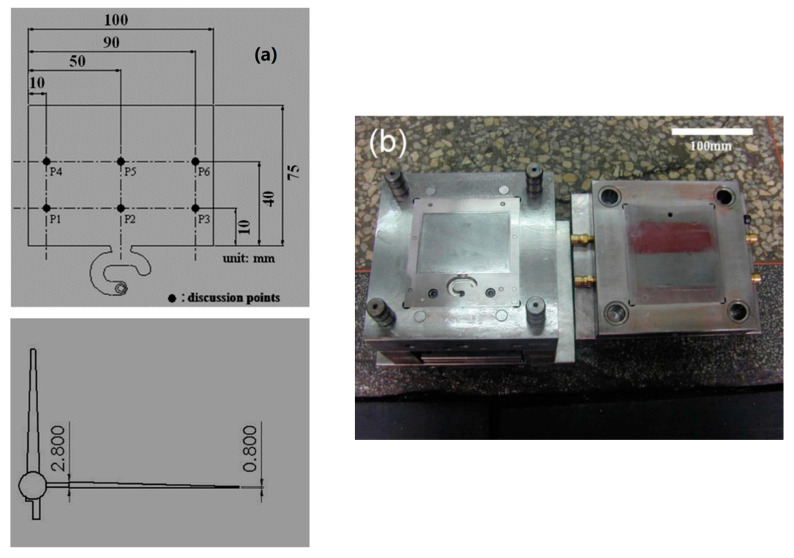
Layout and dimensions on mold. (**a**) Mold cavity and locations of discussion points. (**b**) Mold.

**Figure 2 polymers-14-03077-f002:**
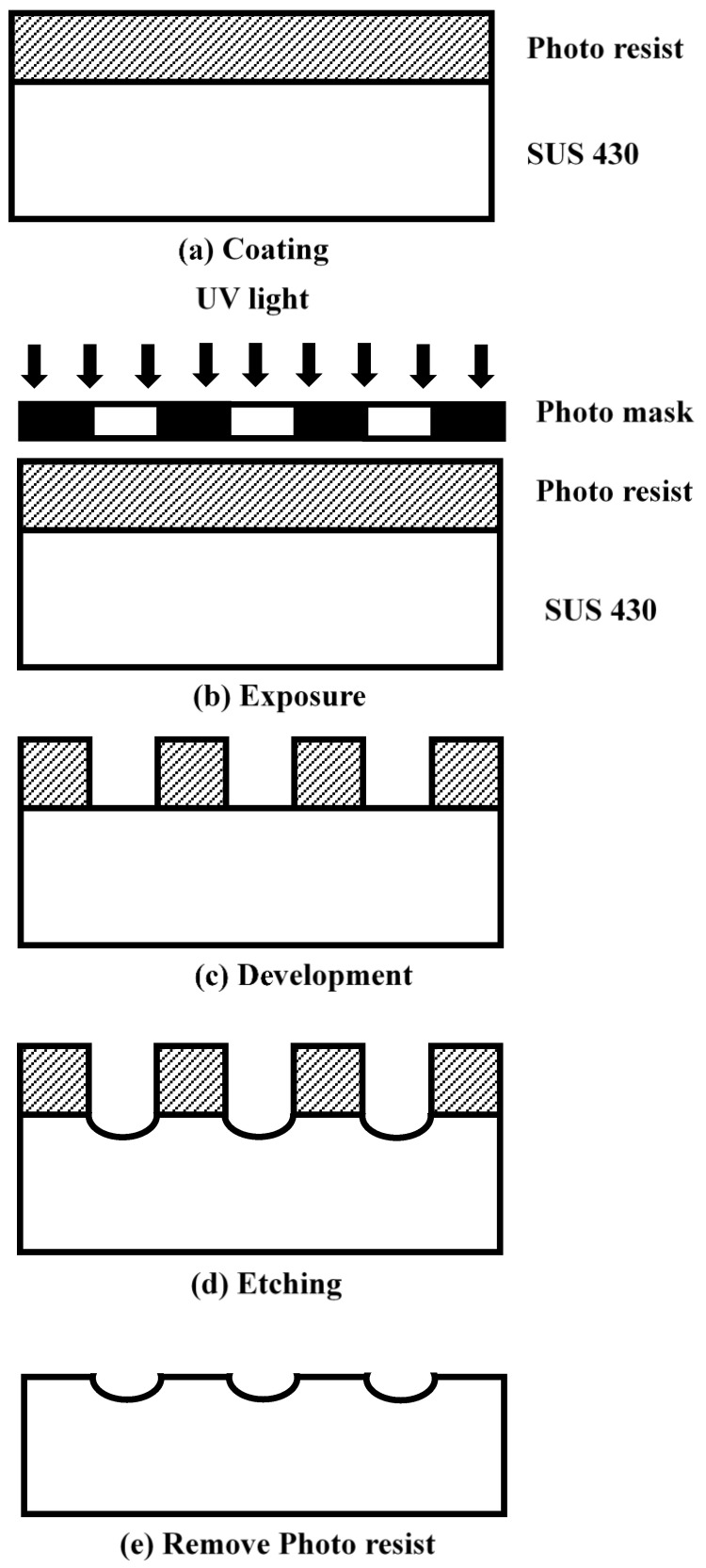
Process chart for photo-etching of mold insert.

**Figure 3 polymers-14-03077-f003:**
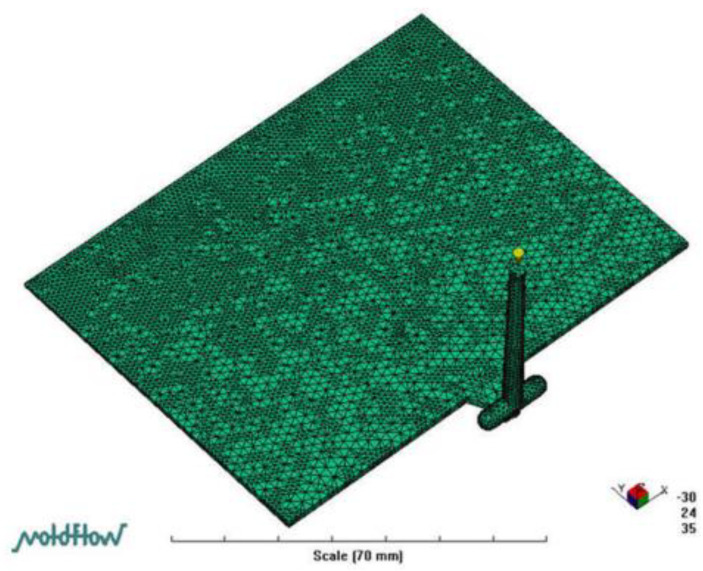
3D-mesh on Moldflow analysis.

**Figure 4 polymers-14-03077-f004:**
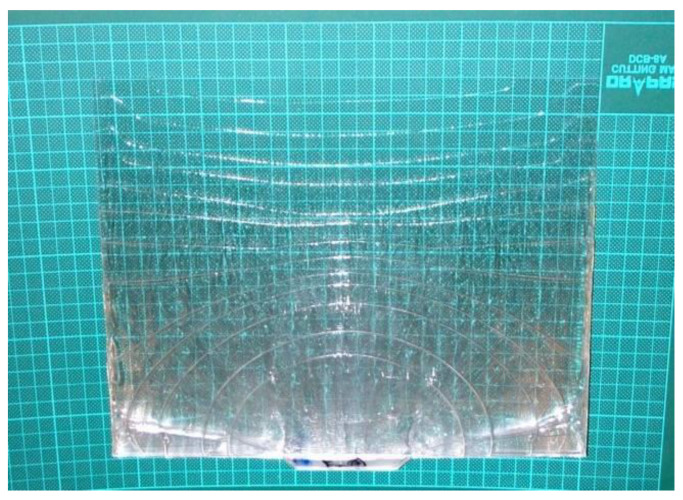
The bad flow situation of light guiding plate on micro-injection molding.

**Figure 5 polymers-14-03077-f005:**
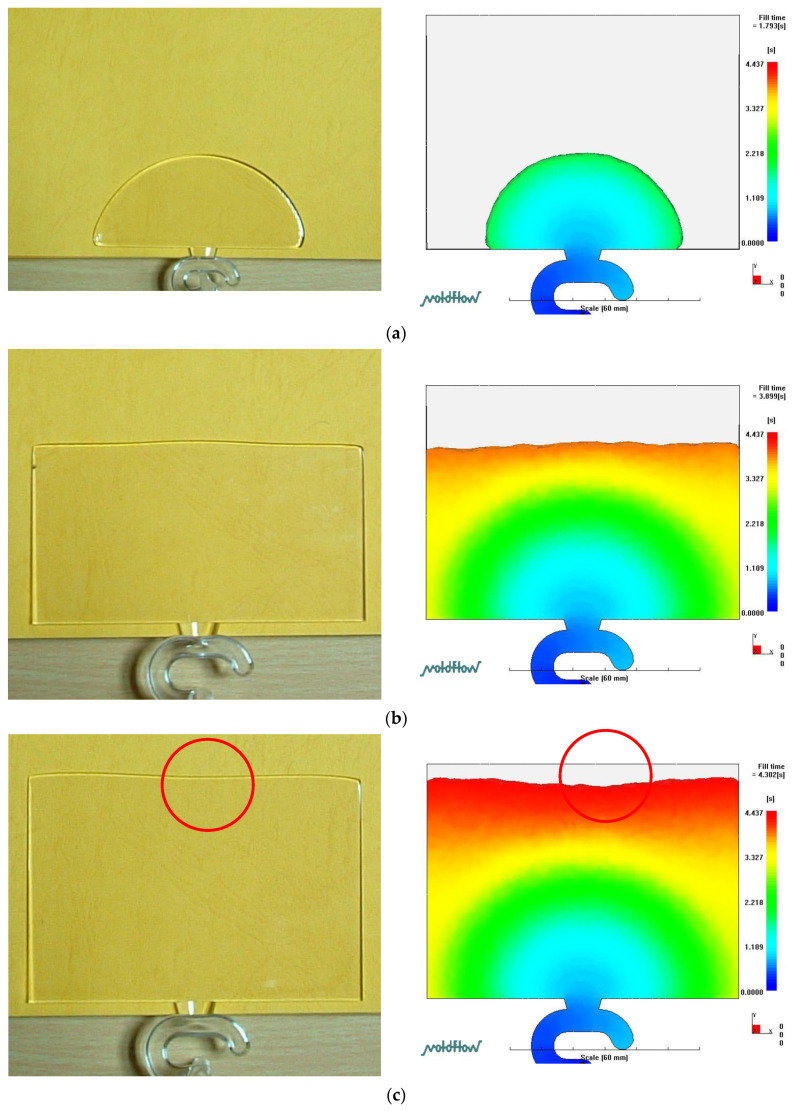
Melt front hysteresis between experiment and 3D numerical simulation. (**a**) Filling phage: 30%, (**b**) filling phage: 60%, (**c**) filling phage: 90%, (**d**) filling phage: 100%.

**Figure 6 polymers-14-03077-f006:**
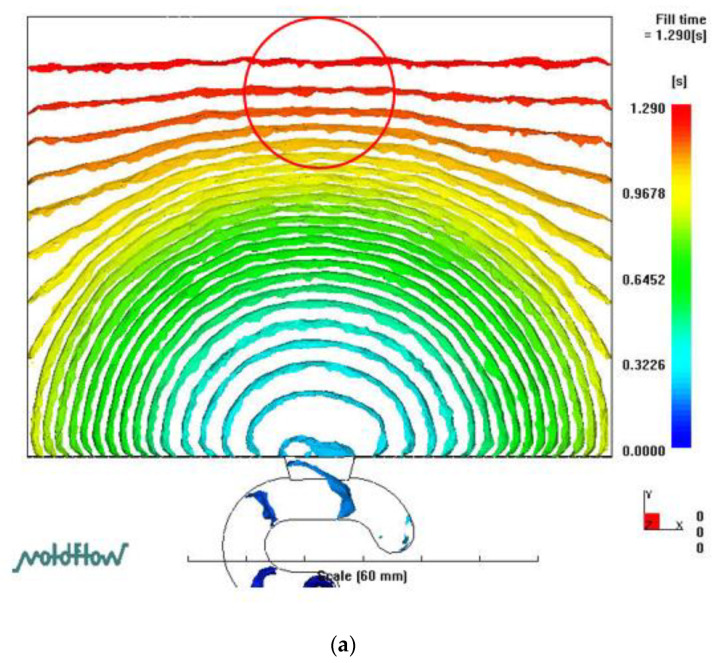
Melt front situation for different injection speeds. (**a**) Melt front uniformity. Mold temp: 60 °C, Melt temp: 245 °C, Injection speed 10 cm/s. (**b**) Melt front hysteresis. Mold temp: 60 °C, Melt temp: 245 °C, Injection speed 3 cm/s.

**Figure 7 polymers-14-03077-f007:**
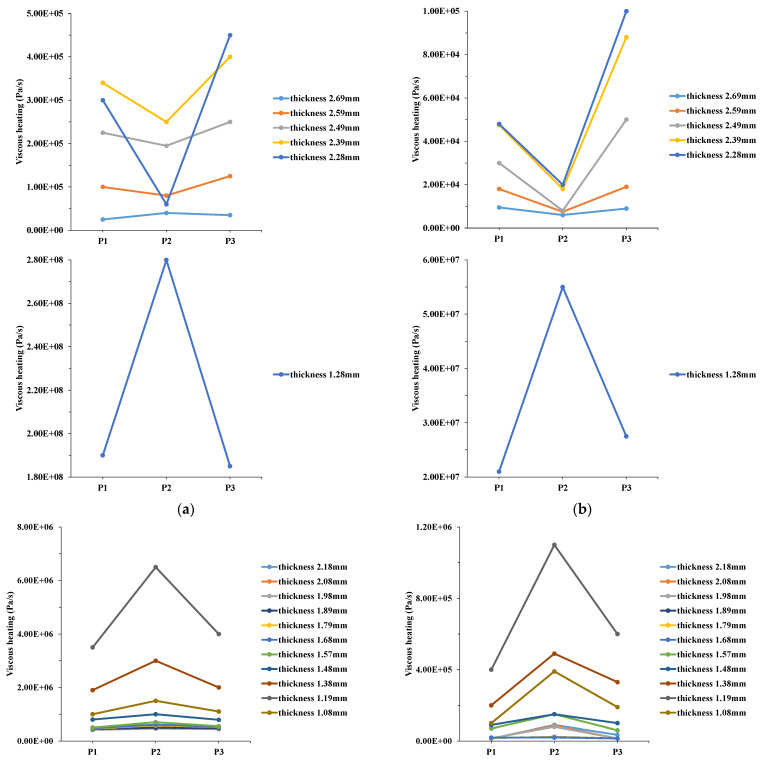
Viscous heating, temperature, and velocity distribution for different positions at 30% filling phage. (**a**) Viscous heating at P1, P2, P3 (melt front uniformity), (**b**) viscous heating at P1, P2, P3 (melt front hysteresis), (**c**) viscous heating at P1, P2, P3 (melt front uniformity), (**d**) viscous heating at P1, P2, P3 (melt front hysteresis), (**e**) temperature and velocity distribution at P1, P2, P3 (melt front uniformity), (**f**) temperature and velocity distribution at P1, P2, P3 (melt front hysteresis).

**Figure 8 polymers-14-03077-f008:**
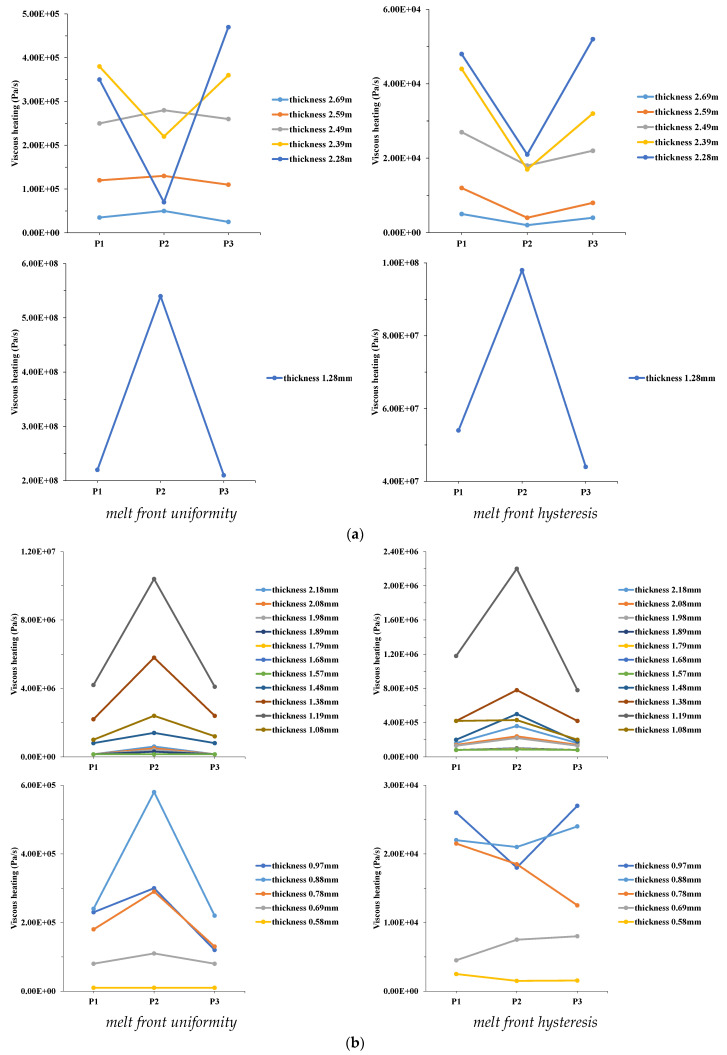
Viscous heating, temperature, and velocity distribution for different positions at 60% filling phage. (**a**) Viscous heating at P1, P2, P3. (**b**) Viscous heating at P1, P2, P3. (**c**) Viscous heating at P4, P5, P6. (**d**) Temperature distribution at P1–P6. (**e**) Velocity distribution at P1–P6.

**Figure 9 polymers-14-03077-f009:**
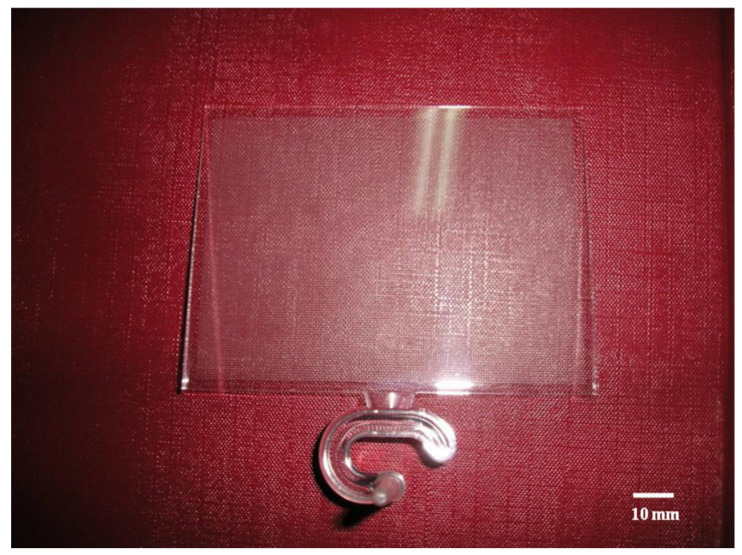
The molded light guiding plate.

**Table 1 polymers-14-03077-t001:** The processing parameters and values in the experiments.

Level Processing Parameters	1	2	3
A. Mold temperature (℃)	50	60	70
B. Melt temperature (℃)	240	245	250
C. Injection speed (cm/s)	3	6	10

## Data Availability

All of the data used to support the findings of this study are included within the article.

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
