# Peer review of "Analysis of Melt Front Behavior of a Light Guiding Plate during the Filling Phase of Micro-Injection Molding"

_polymers, 2022, doi:10.3390/polym14153077_

Round 1

Reviewer 1 Report

The hysteresis phenomenon at melt front in micro-injection molding process of light guide plate  is investigated by 3D numerical simulations from viscosity heat generation, temperature distribution and velocity distribution. This problem is then solved by increasing the injection speed. A few specific comments:

1. The authors considered only one factor as injection speed, two levels of the 3cm/s and 10cm/s in the study. Then the conclusion that the hysteresis phenomenon was solved is not rigorous enough.

2.  In the abstract, the causes of the hysteresis phenomenon, the solutions and the final results are not involved.

3. In the abstract, the authors use the term MM, but it was not defined in the paper.

4. Line 144: The authors said A theoretical analysis through fluid mechanics and heat transfer was applied to explain the hysteresis and ……, In practice, the authors did not use heat transfer mechanism to explain the hysteresis phenomenon.

5. Fig. 1(c) is not necessary.

6. Line 182: We Judged of two characteristics of the flow characteristics: melt front uniformity and melt front hysteresis…….I think these two parameters describe a phenomenon, whether the melt front can reach the end position at the same time.

7. Line 188: In table 2, the the authors list three factors, namely mold temperature, melt temperature and injection speed, at levels 1, 2 and 3. However, only "the mold temperature of 60℃, melt temperature of 245 and injection speed of 3 cm/s" and "the mold temperature of 60℃, melt temperature of 245 and injection speed of 10 cm/s" conditions are considered in this study.

8. In Fig.3, the 3D mesh should be partially enlarged, otherwise details will not be visible.

9. In Fig.4, how to obtain the position of the polymer melt front of the LGP, should be briefly introduced.

10. In Fig. 7 and Fig. 8, there is not a one-to-one correspondence between figure and number, and some figure descriptions are inconsistent with figure content, such as figure 7 (d).

Author Response

        July 15, 2022

Professor and Chairman Y. K. Shen

Taipei Medical University

250 Wu-Xin Street, Taipei City, Taiwan,

R.O.C.110

Tel:886-2-27361661 ext. 5147

Fax: 886-2-27362295

E-mail: ykshen@tmu.edu.tw

Dear Prof. Dr. Ming-Shyan Huang,

Enclosed please find the revised manuscript of the paper entitled “Analysis of melt front behavior of a light guiding plate during the filling phase of micro-injection molding” to submit to Polymers, special issue: Frontiers in Injection Molding of Polymers. These changes and additions in the revised manuscript are shown red color. If you have any problem, please contact with me as soon as possible.

The following items are the reviewer’s comments and the author’s replay.

Reviewer #1:

The hysteresis phenomenon at melt front in micro-injection molding process of light guide plate is investigated by 3D numerical simulations from viscosity heat generation, temperature distribution and velocity distribution. This problem is then solved by increasing the injection speed. A few specific comments:

.

  1. The authors considered only one factor as injection speed, two levels of the 3cm/s and 10cm/s in the study. Then the conclusion that the hysteresis phenomenon was solved is not rigorous enough.

Answer: Thanks the reviewer’s opinions. The authors’ laboratory has many years of experience. The injection speed is very important for the molding process of the large-scale light guiding plate with microstructure at micro-injection molding. Because the size of the light guiding plate becomes larger on today, in order to reduce its weight, the thickness direction of light guiding plate is designed as a wedge-shaped plate, and the injection speed will affect the position of the melt front at filling phase of micro-injection molding. It is very important to let the plastic melt front reach the end of the mold cavity at the same time on filling phase of micro-injection molding. The injection speed value used is the laboratory experience, and other molding parameters such as mold temperature and melt temperature are secondary parameters. Therefore, this study focuses on the injection speed, and finds to change the position of the plastic melt front through the slower or faster injection speed on micro-injection molding.

  1. In the abstract, the causes of the hysteresis phenomenon, the solutions and the final results are not involved.

Answer: Thanks the reviewer’s opinions. The authors had added the hysteresis phenomenon, the solutions and the final results to the Abstract (Page 1).

  1. In the abstract, the authors use the term “MM”, but it was not defined in the paper.

Answer: Thanks the reviewer’s opinions. The author hereby apologizes for mistakenly planting MIM as MM, and the correct text is MIM (micro-injection molding) (Page 1).

  1. Line 144: The authors said “A theoretical analysis through fluid mechanics and heat transfer was applied to explain the hysteresis and ……,” In practice, the authors did not use heat transfer mechanism to explain the hysteresis phenomenon.

Answer: Thanks the reviewer’s opinions. In the theoretical analysis part, the authors combine continuity equation, momentum equation and energy equation together, and then applies the flow mechanics and heat transfer analysis to explain the melt front hysteresis or uniformity. The author also uses the viscous heating term () in the energy equation to explain the aforementioned phenomenon in detail.

  1. Fig. 1(c) is not necessary.

Answer: Thanks the reviewer’s opinions. The authors have removed Fig. 1(c).

  1. Line 182: “We Judged of two characteristics of the flow characteristics: melt front uniformity and melt front hysteresis…….”I think these two parameters describe a phenomenon, whether the melt front can reach the end position at the same time.

Answer: Thanks the reviewer’s opinions. Yes, these two parameters describe a phenomenon, whether the melt front can reach the end position of mold cavity at the same time on micro-injection molding. The authors also indicated the situation for the flow characteristics in the Abstract. The sentences are “During the filling phase of MIM, the enter polymer melt front of the LGP should reach the end of the mold cavity at the same time” (Page 1, Line 12-13).

  1. Line 188: In table 2, the the authors list three factors, namely mold temperature, melt temperature and injection speed, at levels 1, 2 and 3. However, only "the mold temperature of 60℃, melt temperature of 245℃ and injection speed of 3 cm/s" and "the mold temperature of 60℃, melt temperature of 245℃ and injection speed of 10 cm/s" conditions are considered in this study.

Answer: Thanks the reviewer’s opinions. Injection speed is a very important factor for the molding process of large light guiding plates with microstructures at micro-injection molding on the authors’ experiences. Due to the light guiding plate of larger size, in order to reduce its weight, the thickness direction of the light guiding plate must be designed as a wedge-shaped plate, and the injection speed will affect the position of the melt front at the filling phase of micro-injection molding. It is important to have the plastic melt front reach the end of the cavity at the same time on micro-injection molding. The injection speed values used are laboratory experience. The other molding parameters are secondary process parameters such as mold temperature of 60℃ and melt temperature of 245℃. Therefore, this study focuses on the injection speed and how the position of the plastic melt front can be changed by a slower or faster injection speed (3 cm/s, 10 cm/s) at micro-injection molding.

  1. In Fig.3, the 3D mesh should be partially enlarged, otherwise details will not be visible.

Answer: Thanks the reviewer’s opinions. The authors have enlarged in on the 3D mesh of Figure 3 (Page 8).

  1. In Fig.4, how to obtain the position of the polymer melt front of the LGP, should be briefly introduced.

Answer: Thanks the reviewer’s opinions. The authors use the six-melt front discussion points (from P1 to P6) in Figure 1 to discuss the melt front position. In Figure 4, the authors control the shape of the plastic melt front in the entire filling phase with different injection times during the filling process of the light guiding plate on micro-injection molding. Therefore, the melt fronts of different shapes on the light guiding plate can see in Figure 4. Placing the light guiding plate on the checkered plate can confirm the melt front positions of the six discussion points in Figure 1. The situation can see from Figure 4 that the melt front shows hysteresis, which is not conducive to the molding process. This study will discuss how to improve this situation at later in this study (Page 8).

  1. In Fig. 7 and Fig. 8, there is not a one-to-one correspondence between figure and number, and some figure descriptions are inconsistent with figure content, such as figure 7 (d).

Answer: Thanks the reviewer’s opinions. The authors have revised the corresponding relationship between the figure and the number in Figure 7 and Figure 8. The authors also revised the relationship with the article descriptions (Page 12-22).

Thank you for your review and processing.

                                          Sincerely yours,

Yung-Kang Shen

Professor and Chairman

School of Dental Technology

Taipei Medical University

Reviewer 2 Report

The article discusses the design of a novel gate with a focus on micro-injection process optimization to develop the light guiding plate (LGP) using PMMA. The manuscript lacks the discussion on the polymer part such as the effect of variation in polymer properties on the observed micro-injection performance etc., but rather focuses on the process development only. 

To attract the correct readership, I suggest authors to submit the manuscript to a more relevant journals from MDPI such as "Micromachines" or "Processes".  

Author Response

        July 15, 2022

Professor and Chairman Y. K. Shen

Taipei Medical University

250 Wu-Xin Street, Taipei City, Taiwan,

R.O.C.110

Tel:886-2-27361661 ext. 5147

Fax: 886-2-27362295

E-mail: ykshen@tmu.edu.tw

Dear Prof. Dr. Ming-Shyan Huang,

Enclosed please find the revised manuscript of the paper entitled “Analysis of melt front behavior of a light guiding plate during the filling phase of micro-injection molding” to submit to Polymers, special issue: Frontiers in Injection Molding of Polymers. These changes and additions in the revised manuscript are shown red color. If you have any problem, please contact with me as soon as possible.

The following items are the reviewer’s comments and the author’s replay.

Reviewer #2:

  1. The article discusses the design of a novel gate with a focus on micro-injection process optimization to develop the light guiding plate (LGP) using PMMA. The manuscript lacks the discussion on the polymer part such as the effect of variation in polymer properties on the observed micro-injection performance etc., but rather focuses on the process development only.

Answer: Thanks the reviewer’s opinions. Indeed, the properties of the plastic will affect the plastic product during injection molding, such as the amorphous or crystalline properties of the plastic will affect the characteristics of the finished product. The subject of this study is the fabrication of light guiding plate by micro-injection molding, the material of which is PMMA in the market. Therefore, this study instead of focusing on the properties of the plastic, it focuses on the influence of the process parameters of injection molding on the melt front position of the light guiding plate during filling stage of injection molding.

  1. To attract the correct readership, I suggest authors to submit the manuscript to a more relevant journals from MDPI such as "Micromachines" or "Processes".

Answer: Thanks the reviewer’s opinions. The authors still feel that this study is suitable for the journal “Polymers”.

Thank you for your review and processing.

                                          Sincerely yours,

Yung-Kang Shen

Professor and Chairman

School of Dental Technology

Taipei Medical University

Reviewer 3 Report

“Analysis of melt front behavior of a light guiding plate during

the filling phase of micro-injection molding “proposes to overcome defects in light guiding plate production with the support of simulation indications.

Revision or comment

The work is interesting and in a very wide and useful applications of the injection molding process. However, some improvements can be implemented to better enhance its research contribution. Authors sustain that a novel gate was designed but a fan-gate is a classical design, which are its new characteristics in comparison to previous used gate for this kind of application?

Simulation support to correctly individuate injection velocity to obtain a melt front uniformity and to explain it by viscous heating consideration. Why were only three injection values considered? It is not clear if all the combinations of melt and mold temperature and injection velocity were simulated.

Looking at figures 7 and 8, is not clear to what the variation of thickness is due if the test points are fixed by the geometry and the different filling phase. Are these points distributed along a section?

The results section is too long in describing details that are not useful to the all comprehension. In the last part of this section a final molded component is shown but to which process parameters corresponds? How was performed the optimization? Should a reader evaluate them from the previous plots?

I suggest to better organize the paper looking at the key points performed and that are the added value of the works.

Author Response

        July 15, 2022

Professor and Chairman Y. K. Shen

Taipei Medical University

250 Wu-Xin Street, Taipei City, Taiwan,

R.O.C.110

Tel:886-2-27361661 ext. 5147

Fax: 886-2-27362295

E-mail: ykshen@tmu.edu.tw

Dear Prof. Dr. Ming-Shyan Huang,

Enclosed please find the revised manuscript of the paper entitled “Analysis of melt front behavior of a light guiding plate during the filling phase of micro-injection molding” to submit to Polymers, special issue: Frontiers in Injection Molding of Polymers. These changes and additions in the revised manuscript are shown red color. If you have any problem, please contact with me as soon as possible.

The following items are the reviewer’s comments and the author’s replay.

Reviewer #3:

  1. The work is interesting and in a very wide and useful applications of the injection molding process. However, some improvements can be implemented to better enhance its research contribution. Authors sustain that a novel gate was designed but a fan-gate is a classical design, which are its new characteristics in comparison to previous used gate for this kind of application?

Answer: Thanks the reviewer’s opinions. There are many kinds of gates on injection molding, including pine gate, side gate, film gate, ring gate, sprue gate, over lap gate, fan gate, diagram gate, submarine gate, plug gate, banana gate, etc. Because of the large size of the light guiding plate that causes it very difficult to molding success on micro-injection molding, it is important to choose a suitable gate. In this study, the designed curved runner (runner system) combines with the fan gate to do micro-injection molding. This situation causes the melt front will not be too fast at the center of mold cavity during the filling phase just begin, and the injection speed can be increased so that the melt front can reach the end of the mold cavity at the same time on filling phase of micro-injection molding (Figure 5, 6).

  1. Simulation support to correctly individuate injection velocity to obtain a melt front uniformity and to explain it by viscous heating consideration. Why were only three injection values considered? It is not clear if all the combinations of melt and mold temperature and injection velocity were simulated.

Answer: Thanks the reviewer’s opinions. Injection speed is very important factor for the molding process of large light guiding plates with microstructures on the authors’ experiences. Due to the light guiding plate of larger size, in order to reduce its weight, the thickness direction of the light guiding plate must be designed as a wedge-shaped plate. Therefore, the injection speed will affect the position of the melt front at the filling phase of micro-injection molding. It is important to need the plastic melt front to reach the end of the mold cavity at the same time on micro-injection molding. The injection speed values used are laboratory experience and other process parameters are secondary parameters such as mold temperature of 60℃ and melt temperature of 245℃. Therefore, this study focuses on the injection speed and changes the position of the plastic melt front by a slower or faster injection speed (3 cm/s, 6 cm/s, 10 cm/s) on micro-injection molding.

  1. Looking at figures 7 and 8, is not clear to what the variation of thickness is due if the test points are fixed by the geometry and the different filling phase. Are these points distributed along a section?

Answer: Thanks the reviewer’s opinions. Yes, these measurement points distributed along a section of thickness direction. The light guiding plate is a wedge-shaped plate (that is, the thickness direction is inconsistent). Therefore, it is very important to know the temperature value of each measurement point along the thickness direction, because the plastic melt temperature will affect the viscosity of the plastic melt, and its viscosity will affect the advancing speed of the plastic melt front. Because it is impossible to measure the temperature value of thickness direction of each measurement point with an instrument, this study obtains the temperature value of thickness direction of each measurement point by numerical simulation. In this way, according to the aforementioned method, the position of the melt front can change with different injection speeds, so that the melt front can reach the end of the cavity at the same time on micro-injection molding.

  1. The results section is too long in describing details that are not useful to the all comprehension. In the last part of this section a final molded component is shown but to which process parameters corresponds? How was performed the optimization? Should a reader evaluate them from the previous plots?

Answer: Thanks the reviewer’s opinions. The authors have condensed, and commented on the “Results and Discussions” in a simplified and gist style. The authors have presented the values of process parameters of the optimal micro-injection molding for the final light guiding plate product. This allows readers to use the results obtained in this study as their references to fabricate the appropriate light guiding plate by on micro-injection molding (Page 9-22).

  1. I suggest to better organize the paper looking at the key points performed and that are the added value of the works.

Answer: Thanks the reviewer’s opinions. The authors can update the manuscript and make it more mainly the manuscript, and emphasize the main point of the manuscript.

Thank you for your review and processing.

                                          Sincerely yours,

Yung-Kang Shen

Professor and Chairman

School of Dental Technology

Taipei Medical University

Reviewer 4 Report

The article submitted by Lin et al. (polymers-1774073), entitled “Analysis of melt front behavior of a light guiding plate during the filling phase of micro-injection molding”, investigated the flow behavior of a wedge-shaped light guiding plate during micro-injection molding experimentally and numerically. The melt flow front was the focus and the viscous heating was used to explain the experimental results. Overall, the topic is interesting, and it can be classified as a typical application of polymer. However, this article is somewhat ill-organized. For the benefit of the reader, this reviewer would like to suggest a revision before its publication.

1) Some parts of this article should be shortened. In this reviewer’s opinion, the introduction of the photo-etching of the mold insert is not necessary. The table 1, equation (1)~(7), and figure 1(c) can be deleted, since they have actually no much relationship with this article. The so-called “numerical simulation” is only an application of commercial software, MOLDFLOW, and version used was even published in the 10-years before.    

2) While simplifying the content, the influence of the different flow front caused by the different injection parameters, if possibly, is suggested to be discussed more. Especially, the performance (or others) of the fabricated samples could be evaluated, if possibly, and it can improve the significance of this article.

3) The discussion of the numerical simulation is basically macro-scale. For micro-injection molding, the difference of the simulation between the micro-injection molding and conventional injection molding should be stressed. In addition, the micro-scale and/or multi-scale simulation is better to mention for improving the impact of the article. Authors may refer to this article, Journal of Plastic Film and Sheeting, 2016, 32, 239-271, and introduce the dual-scale simulation briefly about this concern.

4) The language should be edited carefully. For example, what is the meaning of the MM (L18, L48, L77, etc,)? In addition, please modifying the types of the lines in figure 8 for making a distinction. In figure 9(a), the P1 and P3 are basically overlapping. Similar situations should be paid attention and revised carefully.   

In conclusion, some revisions are suggested before its publication.

Author Response

        July 15, 2022

Professor and Chairman Y. K. Shen

Taipei Medical University

250 Wu-Xin Street, Taipei City, Taiwan,

R.O.C.110

Tel:886-2-27361661 ext. 5147

Fax: 886-2-27362295

E-mail: ykshen@tmu.edu.tw

Dear Prof. Dr. Ming-Shyan Huang,

Enclosed please find the revised manuscript of the paper entitled “Analysis of melt front behavior of a light guiding plate during the filling phase of micro-injection molding” to submit to Polymers, special issue: Frontiers in Injection Molding of Polymers. These changes and additions in the revised manuscript are shown red color. If you have any problem, please contact with me as soon as possible.

The following items are the reviewer’s comments and the author’s replay.

Reviewer #4:

  1. Some parts of this article should be shortened. In this reviewer’s opinion, the introduction of the photo-etching of the mold insert is not necessary. The table 1, equation (1)~(7), and figure 1(c) can be deleted, since they have actually no much relationship with this article. The so-called “numerical simulation” is only an application of commercial software, MOLDFLOW, and version used was even published in the 10-years before.

Answer: Thanks the reviewer’s opinions. The authors had cancelled the introduction of the photo-etching of the mold insert (Page 4). The authors also deleted the table 1, and figure 1(c) (Page 4, 6). Because the authors use the equation (1)~(7) to explain the change the temperature value of the melt front at the measurement point, especially by equation 5, and therefore the authors retain them.

  1. While simplifying the content, the influence of the different flow front caused by the different injection parameters, if possibly, is suggested to be discussed more. Especially, the performance (or others) of the fabricated samples could be evaluated, if possibly, and it can improve the significance of this article.

Answer: Thanks the reviewer’s opinions. The authors have made amendments to the reviewer's comments. Injection speed is a very important factor for the molding process of large light guiding plates with microstructures on the authors’ experiences. Due to the light guiding plate of larger size, in order to reduce its weight, the thickness direction of the light guiding plate must be designed as a wedge-shaped plate. The injection speed will affect the position of melt front at the filling phase of micro-injection molding. It is important to need the plastic melt front reach the end of the mold cavity at the same time on micro-injection molding. The injection speed values used are laboratory experience and other process parameters are secondary parameters such as mold temperature of 60℃ and melt temperature of 245℃. Therefore, this study focuses on the injection speed and changes the position of the plastic melt front by a slower or faster injection speed (3 cm/s, 6 cm/s, 10 cm/s) on micro-injection molding (Page 12-22). The authors also evaluated the performance (or others) of the fabricated samples (Page 21-22).  

  1. The discussion of the numerical simulation is basically macro-scale. For micro-injection molding, the difference of the simulation between the micro-injection molding and conventional injection molding should be stressed. In addition, the micro-scale and/or multi-scale simulation is better to mention for improving the impact of the article. Authors may refer to this article, Journal of Plastic Film and Sheeting, 2016, 32, 239-271, and introduce the dual-scale simulation briefly about this concern.

Answer: Thanks the reviewer’s opinions. The authors had explained the differences between micro-injection molding and traditional injection molding in Kukla on page 2 (2nd paragraph of 1. Introduction). The authors had added the Journal of Plastic Film and Sheeting (2016, 32, 239-27) to Introduction part (Page 3, Line 21-25; Page 25).

  1. The language should be edited carefully. For example, what is the meaning of the MM (L18, L48, L77, etc,)? In addition, please modifying the types of the lines in figure 8 for making a distinction. In figure 9(a), the P1 and P3 are basically overlapping. Similar situations should be paid attention and revised carefully.

Answer: Thanks the reviewer’s opinions. MM (L18, L48, L77) is a misspelled, the correct word is MIM, the author has corrected them (Page 1-2). The authors had modified the types of the lines in figure 8. The authors also had double checked for all figure epically for figure 8 (Page 16-21).  

  1. In conclusion, some revisions are suggested before its publication.

Answer: Thanks the reviewer’s opinions. The authors had modified the conclusion in the manuscript (Page 22).

Thank you for your review and processing.

                                          Sincerely yours,

Yung-Kang Shen

Professor and Chairman

School of Dental Technology

Taipei Medical University

Round 2

Reviewer 1 Report

I think The manuscript has been sufficiently improved to warrant publication in Polymers.

Reviewer 2 Report

Authors have responded the questions and concerns of the reviewer#2 partially satisfactorily. 

In the response of Q#2 from the Reviewer #2: Authors agreed that the manuscript is primarily discussing the process engineering in the micro-injection molding, rather than the effect of polymer chemistry (or properties). Therefore, the reason for not considering the manuscript transfer to more aligned journals such as MDPI processes or micromachines is not very clear.

Reviewer 3 Report

Authors have attended to reviewer suggestions to improve the clarity and robustness of the paper.

Reviewer 4 Report

The revised version is better than before. I recommend its publication in present form.